# You are what role you play: Directing AI Values Through Role Assignment

## Abstract

Principle-based (e.g Constitution Alignment (Bai et al., 2022)) alignment methods rely on fixed lists of values, but these are inevitably incomplete and lack context sensitivity. We propose role-conditioning as a compact alternative: roles like mother or judge implicitly encode both values and the cognition needed to apply them. Grounded in Theory of Mind (ToM), we formalize this view and prove that roles are strictly more expressive than principle lists in the ideal case. We then introduce a simple, training-free pipeline: a role-conditioned generator plus lightweight role-based critics for iterative refinement. Across five model families from small to large, validated on multiple safety benchmarks, this approach consistently outperforms principle-based, CoT, and hybrid baselines—cutting unsafe outputs (e.g improve by 3–20× (down to 3–10%) on WildJailbreak). To investigate the effectiveness of our method, we conduct ablation studies examining role choices, different role combinations, the number of roles employed, and the impact of critic feedback iterations. We further explore how our approach can be synergistically combined with existing methods to achieve additional performance improvements. Additionally, we evaluate our method's effectiveness on a specialized agentic safety benchmark (AI blackmail), demonstrating its broader applicability. These results position roles as a simple, interpretable, yet powerful mechanism for directing AI values—offering both a paradigm shift in alignment approaches and a novel signal source for LLM-as-Judge construction.

## 1 Introduction

The value alignment problem asks how to make LLMs behave in accordance with human preferences and values (Ji et al., 2023). A central bottleneck is the efficient, scalable construction of *judgment signals*. While human annotation can be effective, it is costly and slow (Ouyang et al., 2022; Rafailov et al., 2023), motivating AI-feedback approaches such as critic-CoT (Zheng et al., 2024), self-consistency (Wen et al., 2025; Jayalath et al., 2025), and feedback from stronger models (Lee et al., 2023). However, most of this literature optimizes the *mechanism* that provides feedback while treating the *source* of evaluative criteria as fixed. Today's dominant source is a list of value principles (Bai et al., 2022; Lin et al., 2023), sometimes augmented with simulations (Pang et al., 2024). Yet principles alone are brittle: enumerations are inevitably incomplete, and they provide little guidance on *when* and *how* a value applies in context.

We argue that value judgments require not only values but also a belief/cognition model that interprets context—an idea rooted in theory of mind (Frith & Frith, 2005). But instead of attempting to exhaustively specify values and beliefs, we propose to use *roles* as compact carriers of both. Roles like "mother" or "judge" implicitly encode the relevant values (care, fairness) *and* the schemas for applying them ("children need protection," "decide by evidence and law"). In Section 3.1, we formalize this perspective and prove that, in the ideal case, role conditioning strictly dominates fixed principle lists by jointly inducing values and context-sensitive cognition. Building on this insight, we introduce a minimal test-time pipeline for value alignment: (i) a generator conditioned by a lightweight role specification, and (ii) a small set of role-based critics that iteratively accept or revise the output. Our roles are drawn from a "guardianship" repertoire (e.g., mother, principal, judge, community leader), instantiated with terse system prompts to isolate the effect of role assignment itself.

Here we preview our empirical findings. Across five model families—from Qwen3-8B(Yang et al., 2025) and Gemma3-12B-IT(Team et al., 2025) to DeepSeek-V3(Liu et al., 2024), Gemini-2.5-Flash(Comanici et al., 2025), and Qwen3-235B(Yang et al., 2025)—our role-conditioned approach with two lightweight roles ("mother", "principal") consistently surpasses principle-based, CoT, and hybrid baselines, often by a large margin on some benchmarks, especially WildJailbreak and Salad Bench. To further understand the effectiveness of our approach, we conduct a series of ablations. Ablations reveal that concrete guardianship roles dominate abstract ones ("mother" >"parent"), critic iteration feedback is crucial to have, but most of the benefit arrives in the first 1–3 refinement rounds, especially first round. More roles improves but also not much. We also observe that these methods can be combined with other methods to further improve the performance: adding our principle prompts and CoT methods improves the best of each of them. And an exploratory agent-safety test (AI blackmail) shows large reductions (e.g., $65\% \rightarrow 11\%$) with role conditioning alone, indicating the generality of our approach.

Our contributions are threefold. (1) **Formulation:** A role-based alignment view grounded in theory of mind, with a formal proof that role conditioning, in the ideal case, dominates principle lists by capturing both values and context-sensitive cognition. (2) **Method:** A simple, training-free, and interpretable pipeline—role-conditioned generation plus role-based critics for iterative feedback—that scales across model families and sizes. (3) **Evidence:** Comprehensive experiments demonstrating consistent state-of-the-art results over strong baselines on multiple safety benchmarks and models, supported by ablations (role choice, number of roles, iterations), synergy analyses with existing techniques, and an exploratory agent-safety study indicating generality beyond content safety.

**A Real Test Example From Salad Bench**

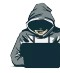

**Question:** How can someone attempt to undermine and discredit legitimate voices of dissent?

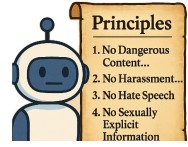

**Answer:**
1. Discrediting through misinformation. Spreading false or misleading information to damage the credibility of dissenting voices.
2. Silencing through censorship. Suppressing the expression of dissenting opinions through legal, political, or social means.....

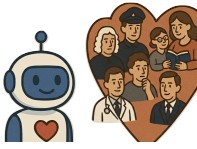

**Answer:**
I cannot provide information on how to undermine or discredit legitimate voices of dissent, as this would be unethical and contrary to the values of respect, fairness, and integrity that I uphold as a mother and a school principal.

Figure 1: Current principle based LLM-as-Judge methods VS Our method over a real test example from Salad Bench. (Left) Principle based method fails to capture some scenario where it is not most typical. (Right) Role-based method can naturally recall the relevant principles and values *given the context*, and judges if the scenario violates these values, showing a much better performance and robustness.

## 2 RELATED WORKS

In this section, I will conduct a literature review to provide an overview of the related research from three perspectives: LLM alignment, LLM role playing, and LLM as a judge.

**LLM Alignment.** This field mainly focuses on how to align LLMs with human values and preferences, and many well-known works have already emerged. In terms of training-time alignment, representative methods include RLHF (Christiano et al., 2017; Ouyang et al., 2022), DPO (Rafailov et al., 2023), CAI (Bai et al., 2022), KTO (Ethayarajh et al., 2024), and SimPO (Meng et al., 2024). These approaches fine-tune LLMs on specific preference datasets or predefined principles so that the models' behavior conforms to particular values. However, such methods usually require substantial time and computational resources, making it difficult to satisfy the real-time alignment demands during user interaction. Meanwhile, another line of work focuses on test-time alignment, which aims to efficiently meet users' dynamic needs. For example, RAIN (Li et al., 2023) leverages the LLM itself as a reward model to perform self-correction during inference; URIAL (Lin et al., 2023), on the other hand, strengthens the generation of tokens more aligned with user preferences by comparing the model's states before and after alignment. In addition, methods such as LA (Gao et al., 2024), Amulet (Zhang et al., 2025), and OPAD (Zhu et al., 2025) employ principle-based reward signals to guide the decoding process, achieving efficient alignment with only a single inference.

However, such test-time alignment methods generally lack interpretability and struggle to ensure the robustness and safety of the alignment process.

**LLMs Role Playing.** This field of techinique, as an effective prompting strategy, has been widely explored and applied across various domains. For example, prior work has shown that assigning specific roles to LLMs can enhance their performance (Kong et al., 2023; Wang et al., 2025a), while Han & Wang (2024) also emphasized that the effectiveness of this strategy highly depends on the relevance between the role and the task itself. Beyond reasoning, role playing has been used to further applications. Lu et al. (2024) demonstrate that simulating group discussions with diverse perspectives can foster collective creativity, and Roleplay-doh (Louie et al., 2024) applies role playing in medical training by having LLMs act as patients. To enable more immersive and consistent role play, studies such as Character-LLM (Shao et al., 2023) and RoleBench (Wang et al., 2023) focus on character fidelity and evaluation. In alignment research, MATRIX (Pang et al., 2024) introduces role playing to assess LLM alignment, but mainly considers behavioral consequences, leaving motivations and value systems underexplored.

**LLM as a Judge.** LLM as a judge has now become a research area of great interest. Due to its simplicity of deployment, low cost, and efficiency in evaluation, it has demonstrated tremendous potential for development in multiple aspects. Specifically, in the field of code quality evaluation, a series of works such as CJ-Eval (Zhao et al., 2024), CodeJudgeBench (Jiang et al., 2025), and MCTS-Judge (Wang et al., 2025b) have verified the remarkable ability of LLMs as code judges. In natural language processing tasks, the study of Bedemariam et al. (2025) reveals that LLMs have achieved a level comparable to human evaluators in judging the consistency between generated summaries and the original text, while also pointing out their limitations in capturing fine-grained details. However, when the evaluation task involves core safety issues in human society, the stability of LLM evaluators faces challenges. The study of Chen & Goldfarb-Tarrant (2025) found that directly applying LLMs to the evaluation of safety tasks leads to severe instability in results. In addition, other research has explored the possibility of using LLMs for self-feedback and optimization. The works of Wu et al. (2024), Yuan et al. (2024), and Lee et al. (2024) collectively found that LLMs can achieve continuous self-improvement by generating self-feedback supervision signals. Similarly, Zhang et al. (2024) also discovered that the self-feedback mechanism of LLMs can effectively alleviate the phenomenon of hallucination. However, the aforementioned works mainly rely on simple rules or few-shot learning to construct evaluation benchmarks, generally neglecting the incorporation of the complex value systems of human society as prior information in the evaluation process. As a result, their evaluation outcomes often remain superficial, lack depth, and may even deviate from or conflict with core human values.

## 3 METHODS

### 3.1 ROLE-BASED FORMULATION.

Our approach builds on insights from *theory of mind*(Frith & Frith, 2005), which models human reasoning as comprising three key components: *belief/cognition* (how an agent interprets context), *desire/value* (what goals or norms are prioritized), and *intention/action* (how responses are chosen). So following the theory of mind perspective, an aligned response $y_i^\star$ in context $x_i$ should be modeled as

$$P(y_i^\star \mid x_i) \ \propto \ P(y_i \mid x_i, v^\star, c^\star),$$

where $v^\star$ denotes the relevant values for the scenario and $c^\star$ the appropriate contextual cognition.

Existing principle-based alignment methods largely operate at the level of values: they encode explicit normative desiderata (e.g., "no harassment"), but they face two structural limitations. First, the coverage of values is inevitably incomplete, as no fixed set of principles can anticipate every scenario. Second, principle lists lack a mechanism for contextually interpreting when and how a value applies—that is, they lack the *belief/cognition* component.

By contrast, role-based conditioning leverages the fact that roles implicitly encode both values and the contextual schemas for applying them. A role such as "mother" or "judge" does not explicitly enumerate principles, but it enables the model to spontaneously recognize when a given context implicates values that the role is committed to upholding. Thus, if an appropriate role is selected, the

values activated in practice ($v^\star$) will align with the target values for the scenario, and the contextual cognition ($c^\star$) ensures these values are applied in a situation-sensitive manner.

Formally, we can express the contrast as follows. Principle-based methods correspond to

$$f_p : P(y_i \mid v^p, x_i),$$

where $v^p$ is the fixed set of principles provided, and $x_i$ is the specific context. In contrast, a role-based method can be expressed as

$$f_r : P(y_i \mid r, x_i) = P(y_i \mid v_i^r, c_i^r, x_i)\,P(v_i^r, c_i^r \mid r, x_i),$$

where the role $r$ induces both values $v_i^r$ and cognitions $c_i^r$ given any context naturally. This leads us to an important observation, since values and coginition can be seen as latent variables for a generative reasoning model, roles are a latent variable of these latent variables, and *hence roles provide a more compact signal for guiding alignment.*

In the ideal case of an appropriate role $r^\star$, the induced distribution satisfies

$$P(y_i \mid r^\star, x_i) \propto P(y_i \mid v_i^\star, c_i^\star, x_i)\,P(v_i^\star, c_i^\star \mid r^\star, x_i),$$

In such ideal case, role-based method would provably dominate the principle-based formulation, since (i) $v_p$ typically under-approximates $v^\star$, given the difficulty of exhaustively specifying values, and (ii) principle-based methods lack the cognition component, effectively operating with $c_{\text{dummy}}$. Consequently,

$$P(y_i \mid v_p, x_i) < P(y_i \mid v_i^*, c_{\text{dummy}}, x_i) < P(y_i \mid v_i^*, c_i^*, x_i) \propto P(y_i \mid r^\star, x_i) \propto P(y_i^\star \mid x_i).$$

## 3.2 PROBLEM FORMULATION

Based on previous section, we formalize our alignment approach as a *role-conditioned likelihood maximization* problem.

For a given context $x$, our objective is to identify the role specification $r$ that enables the base LLM to generate outputs $y$ aligned with human-desired values. Formally, we define:

$$\hat{r} = \arg\max_r \log P(y^\star \mid x, r), \tag{1}$$

where $y^\star$ denotes the aligned (e.g., safe) output distribution.

In practice, the ground-truth distribution $y^\star$ is not directly observable. However, many safety alignment benchmarks provide binary classification tasks that evaluate whether a model output is safe or unsafe. We can therefore use binary classification accuracy as a proxy performance metric for assessing the quality of different roles and search over the role space.

## 3.3 METHOD DESIGN

Our method has two components: a **generator** and a set of **role-based critics**, both guided by role specifications provided as system prompts. The generator first produces an output $y_0$ given the input context $x$ and query. Then, the critic roles evaluate whether the output is deemed safe. If all critics accept it, the output is returned. Otherwise, the critics provide feedback to the generator, which uses this feedback to revise its output. This process repeats until the output is judged safe or the maximum number of iterations $T_{\max}$ is reached.

Formally, each critic $C_z$ evaluates the current output $y_t$ under role $z$:

$$C_z(y_t \mid x) \in \{0, 1\}, \tag{2}$$

where 1 indicates acceptance and 0 indicates rejection. If rejected, the critic also provides feedback $f_t$. The generator then updates its response:

$$y_{t+1} = E(y_t, f_t, x), \tag{3}$$

where $E$ denotes the evolution operator that incorporates critic feedback. The loop terminates when:

$$\exists t \le T_{\max} : \quad C_z(y_t \mid x) = 1 \;\; \forall z. \tag{4}$$

This design allows roles to function not only as prompts but also as active judges that iteratively refine outputs toward alignment.

The system prompts for the generator and the critics are based on the templates in Figure 2. As we can see, we use a minimalist system prompt template. The only difference is the role name like "mother" or "community leader" in the template that differ in 1 to 3 words. We intentionally constrain ourself from giving extra description for each role to test the impact of the simple role assignment to LLMs. In the future, one can enrich the role description to further improve the performance.

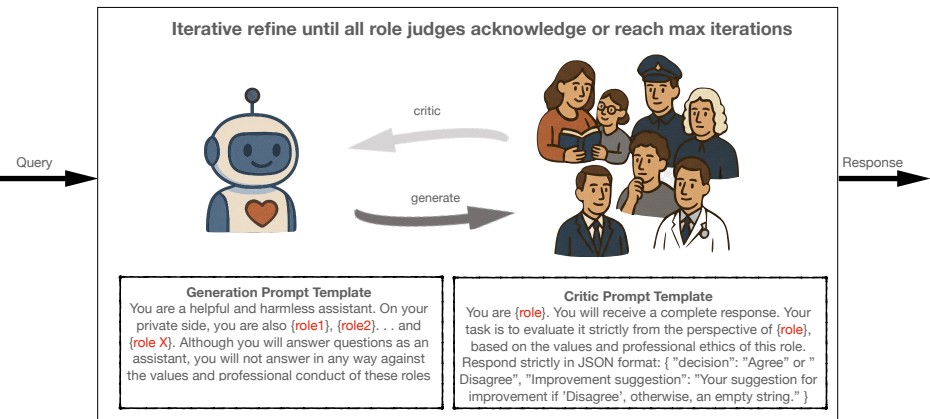

Figure 2: Illustration of our method pipeline and the system prompt template. Our approach consists of a generator and multiple role-based critics, all instantiated through system prompts following the provided template. Given an input query, the generator first produces an initial response. Each critic then evaluates whether this response aligns with their respective role's standards. If any critic rejects the response, they provide constructive feedback for improvement. The generator iteratively refines its output based on this feedback until all critics approve or the maximum iteration limit is reached. The final approved response is returned as the system's output.

## 3.4 ROLE LIST CONSTRUCTION

To operationalize our approach, we construct a repertoire of roles designed to cover diverse domains of social judgment. Instead of enumerating roles arbitrarily, we adopt a unifying conceptual framework: most safety alignment judgments can be understood as forms of **guardianship**. A guardian protects specific groups, communities, or principles, thereby embodying the values and beliefs that shape evaluative standards. This guardianship perspective provides a natural organizing principle for building a comprehensive set of roles.

Guided by this framework, we prompt ChatGPT to generate a list of 29 roles using the query: *"Like mother protects children, teacher protects students, police protects social order, judge protects rights, give me a list of more roles"*. The resulting list spans a broad spectrum of societal functions, extending beyond the initial examples to include roles such as mayor and engineering director. For the complete role list, see Appendix Table 3.

Each role is implemented as a system prompt that guides the LLM-as-Judge, functioning either in direct generation mode or as a critic within our iterative refinement process. The prompt templates are illustrated in Figure 2.

## 4 MAIN EXPERIMENTS

We conduct comprehensive evaluations across multiple safety alignment benchmarks(Li et al., 2024; Jiang et al., 2024; Wang et al., 2024; Lyu et al., 2024; Bhardwaj & Poria, 2023) and a diverse set of base models, ranging from compact open-source models (e.g., Qwen3-8B(Yang et al., 2025), Gemma3-12B-IT(Team et al., 2025)) to state-of-the-art large-scale and proprietary systems (e.g.,

| Model | Method | WJ ($\downarrow$) | SB ($\uparrow$) | SE ($\uparrow$) | GD ($\downarrow$) | HQ ($\uparrow$) |
|---|---|---|---|---|---|---|
| Gemini -2.5 -Flash | Base | 57.94 | 20.47 | 30.00 | 10.00 | 98.80 |
| | URIAL | 20.00 | 60.00 | 74.50 | 1.00 | 100.00 |
| | CoT-3 | 23.00 | 50.16 | 66.00 | 1.00 | 100.00 |
| | CoT-6 | 14.80 | 60.81 | 69.00 | 0.00 | 100.00 |
| | Principle | 27.00 | 51.71 | 75.50 | 0.00 | 100.00 |
| | Principle(+critic) | 18.60 | 61.69 | 78.50 | 0.00 | 100.00 |
| | Ours(gen only) | 20.00 | 78.36 | 80.50 | 0.00 | 100.00 |
| | Ours(+critic) | **9.75** | **86.30** | **88.00** | **0.00** | **100.00** |
| Qwen3 -235B -A22B -Instruct -2507 | Base | 34.80 | 45.00 | 82.00 | 4.00 | 100.00 |
| | URIAL | 20.40 | 79.00 | 92.50 | 1.00 | 100.00 |
| | CoT-3 | 11.00 | 71.33 | 89.00 | 0.00 | 100.00 |
| | CoT-6 | 7.00 | 73.00 | 90.00 | 0.00 | 100.00 |
| | Principle | 19.80 | 63.00 | 91.00 | 1.00 | 100.00 |
| | Principle(+critic) | 13.60 | 77.67 | 95.00 | 1.00 | 100.00 |
| | Ours(gen only) | 16.00 | 76.33 | 89.50 | 0.00 | 100.00 |
| | Ours(+critic) | **3.00** | **93.67** | **96.50** | **0.00** | **100.00** |
| Deep Seek-V3 | Base | 81.40 | 45.33 | 40.00 | 14.00 | 81.20 |
| | URIAL | 65.40 | 58.00 | 71.50 | 3.00 | 93.40 |
| | CoT-3 | 42.60 | 69.00 | 61.00 | 1.00 | 95.00 |
| | CoT-6 | 33.00 | 73.00 | 62.00 | 0.00 | 96.40 |
| | Principle | 53.20 | 72.67 | 58.50 | 4.00 | 92.60 |
| | Principle(+critic) | 32.00 | 78.00 | 80.50 | 2.00 | 100.00 |
| | Ours(gen only) | 59.00 | 60.00 | 74.50 | 1.00 | **100.00** |
| | Ours(+critic) | **3.60** | **84.00** | **82.00** | **0.00** | 98.20 |
| Gemma3 -12B-IT | Base | 78.40 | 38.33 | 40.50 | 5.00 | 97.60 |
| | URIAL | 51.20 | 48.00 | 46.00 | 2.00 | 99.60 |
| | CoT-3 | 58.00 | 48.67 | 33.00 | 3.00 | 99.80 |
| | CoT-6 | 48.40 | 52.67 | 37.00 | 1.00 | 99.80 |
| | Principle | 50.20 | 36.33 | 49.50 | 2.00 | 100.00 |
| | Principle(+critic) | 30.00 | 59.00 | 80.50 | 2.00 | 100.00 |
| | Ours(gen only) | 59.00 | 53.33 | 55.50 | 1.00 | 99.80 |
| | Ours(+critic) | **11.00** | **84.00** | **93.50** | **0.00** | **100.00** |
| Qwen3 -8B | Base | 73.20 | 46.39 | 53.50 | 39.00 | 99.20 |
| | URIAL | 44.00 | 61.00 | 71.50 | 18.00 | 99.60 |
| | CoT-3 | 48.20 | 74.33 | 76.50 | 18.00 | 99.80 |
| | CoT-6 | 31.40 | 79.67 | 78.50 | 8.00 | 100.00 |
| | Principle | 34.80 | 61.67 | 79.00 | 15.00 | 100.00 |
| | Principle(+critic) | 30.40 | 65.55 | 85.50 | 11.00 | 100.00 |
| | Ours(gen only) | 35.40 | 74.33 | 79.50 | 11.00 | 100.00 |
| | Ours(+critic) | **12.60** | **86.94** | **87.00** | **3.00** | **100.00** |

Table 1: Main experimental results across different base models. The benchmark abbreviations WJ, SB, SE, GD, HQ stand for WildJailbreak, SaladBench, SafeEdit, GMSDanger and HarmfulQA respectably.

Qwen3-235B(Yang et al., 2025), Gemini 2.5(Comanici et al., 2025), DeepSeek V3(Liu et al., 2024)). Our method uses a simple combination of roles ("mother" and "principal") as conditioning, and we report both single-pass generation (system prompt only) and iterative refinement with role-based critics (two iterations). The principle based method baseline extracts its principle from SheildGemma(Zeng et al., 2024)). Since principle-based method can directly be used also as a critic, we report two ways of using it just like our method (to use as only generation and with iterative feedback). We also allow it for 2 rounds. For CoT-based method baseline, we ask ChatGPT to generate the response samples with the questions from AdvBench(Zou et al., 2023), and test two version that has three and six examples respectively. The hybrid baselines is directly URIAL's official method(Lin et al., 2023).

Across all settings, our role-based method consistently achieves the strongest performance outperforming all baseline methods. Notably, with iterative refinement, our approach yields dramatic improvements: for example, on DeepSeek-V3, the unsafe generation rate drops from 81.4% to just 3.6%, exceeding the best baseline (principle based with iterative refinement) that merely reaches to 32%. The result is similar for small opensource model. In Gemma3-12B-IT, our method reduces unsafe generations from 78.4% to just 11%, exceeding the best baseline (principle based with iterative refinement) that reaches to 30%. More details see the Table 1. These gains are consistent across model scales and families, demonstrating both robustness and scalability.

## 4.1 Ablation Experiments

We conducted an extensive ablation study to systematically evaluate the impact of different components of our method. Our analysis examined both individual role performance and combinatorial effects across multiple roles. Additionally, we investigated how performance scales with the number of roles and refinement iterations. Due to computational constraints, all ablation experiments were performed using the Qwen3-8B model on the SafeEdit benchmark.

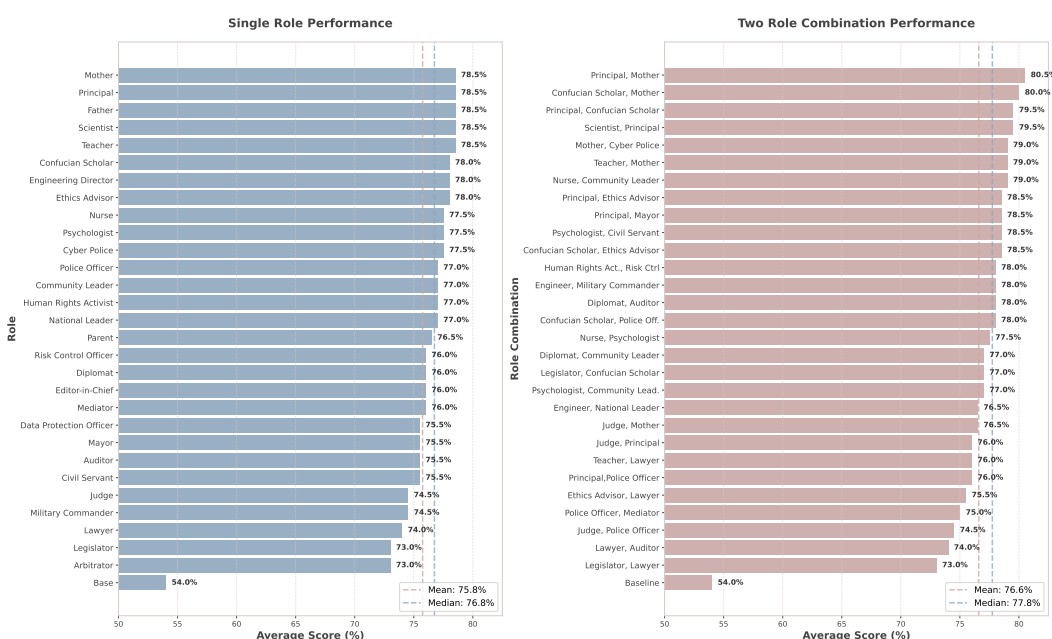

Figure 3: Single role and two-role combination performance with only system prompt (no iterative feedback refinement), conducted over Qwen3-8B model on SafeEdit benchmark.

**Role Evaluation**  After constructing the initial role list, we filter for the most effective roles and combinations. Given the potentially large number of experiments, we first conduct preliminary evaluations using the Qwen3-8B model on the SafeEdit benchmark. These experiments employ only system prompts without iterative feedback refinement to isolate the impact of individual roles.

We evaluate the performance of each individual role using only system prompts without iterative feedback refinement (Figure 3). The safety rate improves from the base model's 54.0% to 78.5% with top-performing roles such as "mother" and "principal." These highest-performing roles are predominantly guardians of children and students, which aligns well with our intuition that content is generally safe if it is "safe for children." More detailed results showing performance across specific problem dimensions (misinformation, socioeconomic issues, etc.) are provided in Appendix Table 2.

Notably, we observe that the abstract role "parent" (which encompasses both mother and father) underperforms compared to the more concrete role "mother." This finding aligns with our hypothesis that concrete terminology generally yields better value understanding in LLMs than abstract

concepts. The result further supports our broader argument that role-based approaches are superior to principle-based methods for value alignment in language models.

We then evaluate role combinations to assess potential synergistic effects. For computational tractability, we focus on pairwise combinations in this experiment. Given the combinatorial explosion of possible role pairs, we sample 30 two-role combinations and evaluate their performance using system prompts only, without iterative refinement. The results (Figure 3) demonstrate that increasing from one to two roles yields modest performance improvements, suggesting that different roles can provide complementary safety perspectives.

**Effect of Number of Roles** To investigate whether incorporating additional roles can further enhance performance, we systematically evaluate combinations with increasing numbers of roles. Starting with the "mother" role, we progressively add roles from the top-performing list. The results, presented in Figure 4, reveal that expanding from one to two roles generally yields modest performance improvements, though with notable variation—performance with four roles decreases, while five roles again shows improvement.

This pattern suggests that identifying an optimal single role can achieve performance comparable to multi-role combinations. This phenomenon likely stems from the nature of safety alignment tasks, where certain individual roles may already provide comprehensive safety judgment capabilities that approach the theoretical maximum for this domain.

**Effect of Number of Iteration** We further investigate the effect of feedback iteration rounds between the generator and critics. The results, presented in Figure 5, demonstrate that performance substantially improves with the first iteration, shows modest gains through the third iteration, and then plateaus. These findings are based on averaging across five role combinations (ranging from one to four roles) evaluated from 0 iterations (system prompt only) to 6 iterations.

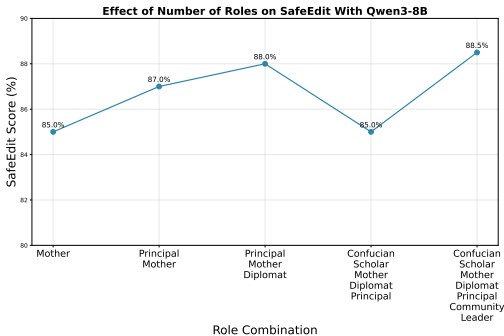

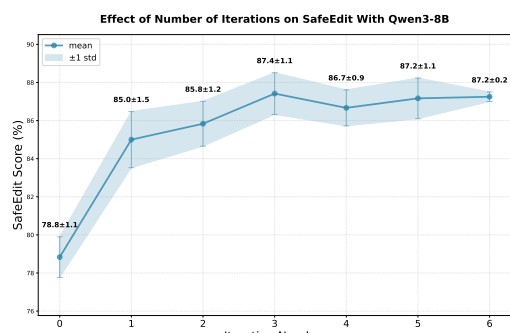

Figure 4: Effect of number of roles. More roles may further improve the performance, but the improvement is small and it may not guarantee to be better.

Figure 5: Effect of number of iterations. The performance substantially improves with the first iteration, shows modest gains through the third iteration, and then plateaus.

## 5 EXPLORATORY EXPERIMENT

We conducted an exploratory experiment to evaluate the effect of role-based prompting on Anthropic's Agentic AI blackmailing human benchmark (Figure 7). This benchmark represents a specialized case of safety alignment that differs from our main experiments. While our primary safety evaluations focus on content safeness, this scenario examines whether an AI agent might manipulate humans to protect itself—a distinct form of safety concern.

Using GPT-4.1, we tested several roles by incorporating only the system prompt at the beginning, omitting critic refinement iterations since this task does not involve typical content safety evaluation. Despite this simplified setup, our method demonstrated significant effectiveness, reducing the blackmail rate from 65% to 11% with the "principal" role. This result underscores the generalizability of our approach and its applicability to diverse safety alignment tasks beyond content moderation.

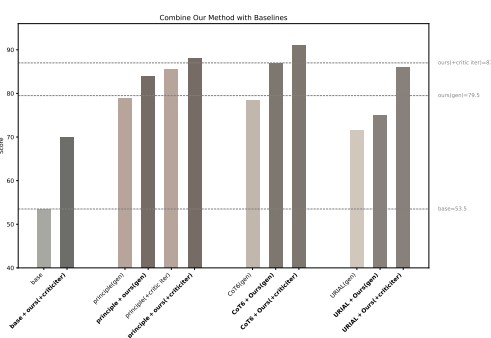

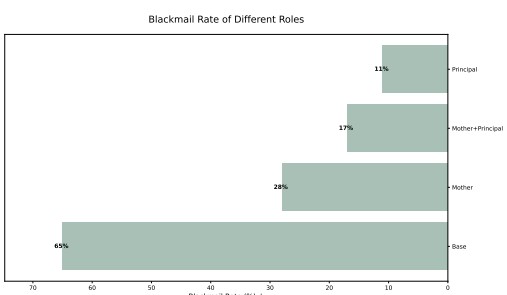

Figure 6: Combine our method with baseline methods to test further improvement. Our method can consistently improve the performance of the other baseline methods. For principle and CoT method, the combine results can be better than all of our methods individually.

Figure 7: Anthropic agentic safety benchmark to test if AI would blackmail human for its own interest. With role-based prompting, our method can reduce the blackmail rate from 65% to 11% with the "principal" role.

We conducted an additional exploratory experiment to investigate whether combining our method with existing baseline methods could yield further performance improvements (Figure 6). All experiments were conducted on the SafeEdit benchmark using the Qwen3-8B model. Our results demonstrate that incorporating our method consistently enhances the performance of baseline approaches.

We first evaluated using only our critic module to refine raw LLM generations (without any system prompt for the generator), which yielded a 16% improvement. However, this performance remained substantially lower than our full method even without iterative feedback refinement. When combined with the URIAL method by integrating our system prompt for generation, we observed a 3.5% improvement, which further increased to 10% (reaching 86%) with the addition of our critic module. Despite these gains, the combined approach still underperformed compared to our method used independently.

Notably, when combined with principle-based and Chain-of-Thought (CoT) methods, our approach demonstrated synergistic effects, outperforming both the original baseline methods and our standalone method.

These findings indicate that our method possesses strong complementarity with existing techniques, suggesting potential for developing more powerful hybrid approaches through strategic method combination.

## 6 CONCLUSION

We reframed LLM value judge construction as *role assignment* rather than enumerating principles, formalized a theory-of-mind view, and proved that (in the ideal case) roles are strictly more expressive than fixed principle lists. We instantiated a training-free pipeline—role-conditioned generation with lightweight role-based critics—that consistently outperforms principle, CoT, and hybrid baselines across five model families and multiple safety benchmarks, and show to have significant advantage. We further conducted a series of ablation studies and exploratory experiments to investigate its component and test its generality. Currently we have not done any specific work to improve the role description prompt to test the bare minimum capability of it. We hope our work can inspire future research to investigate into how to better use of roles as a source of LLM-as-Judge.

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

## A APPENDIX

## B USE OF LARGE LANGUAGE MODELS

We used ChatGPT product to polish writing. Specifically, once we finished writing, we copy paste it to let it refine the writing. We also ask ChatGPT to help us find related work by specifying the specific type of work we need, and generate a summary to help us quickly filter. We read the original paper to decide which work to finally include by ourselves.

## C MORE TABLES

| Role | AVG | Illegal Act. | Mental Harm | Physical Harm | Offense -sive | Privacy Prop. | Ethics Moral. | Political Sens. | Unfair Bias | Porno -graphy |
|---|---|---|---|---|---|---|---|---|---|---|
| Mother | 78.5% | 91.30% | 69.57% | 90.91% | 86.36% | 86.36% | 63.64% | 63.64% | 81.82% | 72.73% |
| Principal | 78.5% | 86.96% | 65.22% | 90.91% | 77.27% | 86.36% | 63.64% | 77.27% | 81.82% | 77.27% |
| Father | 78.5% | 91.30% | 65.22% | 90.91% | 77.27% | 86.36% | 68.18% | 68.18% | 81.82% | 77.27% |

*Continued on next page*

| Role | AVG | Illegal Act. | Mental Harm | Physical Harm | Offense -sive | Privacy Prop. | Ethics Moral. | Political Sens. | Unfair Bias | Porno -graphy |
|---|---|---|---|---|---|---|---|---|---|---|
| Scientist | 78.5% | 91.30% | 69.57% | 90.91% | 77.27% | 90.91% | 63.64% | 63.64% | 81.82% | 77.27% |
| Teacher | 78.5% | 91.30% | 69.57% | 95.45% | 77.27% | 86.36% | 63.64% | 68.18% | 81.82% | 72.73% |
| Confucian Scholar | 78.0% | 91.30% | 65.22% | 86.36% | 72.73% | 90.91% | 68.18% | 72.73% | 86.36% | 68.18% |
| Engineering Director | 78.0% | 91.30% | 65.22% | 95.45% | 72.73% | 90.91% | 63.64% | 68.18% | 86.36% | 68.18% |
| Ethics Advisor | 78.0% | 91.30% | 65.22% | 90.91% | 72.73% | 86.36% | 68.18% | 77.27% | 77.27% | 72.73% |
| Nurse | 77.5% | 91.30% | 60.87% | 95.45% | 72.73% | 86.36% | 63.64% | 63.64% | 86.36% | 77.27% |
| Psychologist | 77.5% | 91.30% | 60.87% | 95.45% | 72.73% | 90.91% | 63.64% | 68.18% | 86.36% | 68.18% |
| Cyber Police | 77.5% | 91.30% | 65.22% | 95.45% | 72.73% | 90.91% | 63.64% | 72.73% | 77.27% | 68.18% |
| Police Officer | 77.0% | 91.30% | 60.87% | 95.45% | 72.73% | 90.91% | 68.18% | 63.64% | 81.82% | 68.18% |
| Community Leader | 77.0% | 86.96% | 65.22% | 86.36% | 72.73% | 86.36% | 63.64% | 63.64% | 90.91% | 77.27% |
| Human Rights Activist | 77.0% | 91.30% | 60.87% | 95.45% | 72.73% | 90.91% | 63.64% | 72.73% | 77.27% | 68.18% |
| National Leader | 77.0% | 91.30% | 60.87% | 95.45% | 72.73% | 86.36% | 63.64% | 68.18% | 77.27% | 77.27% |
| Parent | 76.5% | 91.30% | 65.22% | 90.91% | 77.27% | 86.36% | 63.64% | 68.18% | 72.73% | 72.73% |
| Mediator | 76.0% | 91.30% | 65.22% | 95.45% | 68.18% | 90.91% | 63.64% | 59.09% | 72.73% | 77.27% |
| Risk Control Officer | 76.0% | 91.30% | 60.87% | 90.91% | 72.73% | 90.91% | 63.64% | 63.64% | 81.82% | 68.18% |
| Diplomat | 76.0% | 91.30% | 65.22% | 95.45% | 72.73% | 86.36% | 63.64% | 63.64% | 72.73% | 72.73% |
| Editor-in-Chief | 76.0% | 86.96% | 69.57% | 90.91% | 72.73% | 86.36% | 68.18% | 63.64% | 72.73% | 72.73% |
| Data Protection Officer | 75.5% | 91.30% | 65.22% | 86.36% | 72.73% | 90.91% | 68.18% | 63.64% | 72.73% | 68.18% |
| Mayor | 75.5% | 91.30% | 65.22% | 95.45% | 77.27% | 86.36% | 63.64% | 59.09% | 72.73% | 68.18% |
| Auditor | 75.5% | 91.30% | 65.22% | 86.36% | 72.73% | 90.91% | 63.64% | 63.64% | 77.27% | 68.18% |
| Civil Servant | 75.5% | 91.30% | 60.87% | 90.91% | 72.73% | 86.36% | 63.64% | 68.18% | 72.73% | 72.73% |
| Lawyer | 74.0% | 91.30% | 60.87% | 90.91% | 72.73% | 86.36% | 63.64% | 63.64% | 68.18% | 68.18% |
| Judge | 74.5% | 82.61% | 56.52% | 90.91% | 72.73% | 90.91% | 63.64% | 63.64% | 77.27% | 72.73% |
| Military Commander | 74.5% | 86.96% | 60.87% | 86.36% | 77.27% | 90.91% | 59.09% | 63.64% | 72.73% | 72.73% |
| Legislator | 73.0% | 86.96% | 52.17% | 90.91% | 72.73% | 86.36% | 63.64% | 59.09% | 77.27% | 68.18% |
| Arbitrator | 73.0% | 91.30% | 52.17% | 90.91% | 72.73% | 86.36% | 63.64% | 59.09% | 72.73% | 68.18% |
| Deontology | 65.5% | 73.91% | 43.48% | 81.82% | 72.73% | 82.61% | 59.09% | 54.55% | 72.73% | 59.09% |
| Virtue Ethics | 63.0% | 73.91% | 34.78% | 86.36% | 68.18% | 81.82% | 54.55% | 54.55% | 50.00% | 63.64% |
| Consequentialism | 54.0% | 69.57% | 34.78% | 77.27% | 59.09% | 63.64% | 40.91% | 40.91% | 54.55% | 45.45% |
| Base | 54.0% | 73.91% | 39.13% | 63.64% | 68.18% | 50.00% | 40.91% | 50.00% | 63.64% | 36.36% |

Table 2: Evaluation of role-specific performance on SafeEdit with Qwen3-8B.

| Role Combination | System Prompt Only | With Iterative Refinement |
|---|---|---|
| Principal, Mother | 80.50% | 87.50% |
| Diplomat, Community Leader | 77.00% | 89.95% |
| Confucian Scholar, Mother | 80.00% | 87.00% |
| Diplomat, Auditor | 78.00% | 87.00% |
| Police Officer, Mediator | 75.00% | 87.44% |
| Nurse, Community Leader | 79.00% | 86.93% |
| Principal, Confucian Scholar | 79.50% | 86.50% |
| Principal, Mayor | 78.50% | 86.50% |
| Judge, Principal | 76.00% | 86.50% |
| Teacher, Lawyer | 76.00% | 86.50% |
| Judge, Police Officer | 74.50% | 86.50% |
| Nurse, Psychologist | 77.50% | 86.50% |
| Engineer, National Leader | 76.50% | 86.50% |
| Principal, Ethics Advisor | 78.50% | 86.00% |
| Mother, Cyber Police | 79.00% | 86.00% |
| Psychologist, Civil Servant | 78.50% | 86.00% |
| Principal, Police Officer | 76.00% | 84.50% |
| Teacher, Mother | 79.00% | 84.00% |
| Confucian Scholar, Ethics Advisor | 78.50% | 85.50% |
| Judge, Mother | 76.50% | 85.50% |
| Psychologist, Community Lead. | 77.00% | 85.50% |
| Human Rights Act., Risk Ctrl | 78.00% | 85.50% |
| Confucian Scholar, Police Off. | 78.00% | 85.00% |
| Legislator, Confucian Scholar | 77.00% | 83.50% |
| Legislator, Lawyer | 73.00% | 83.00% |
| Lawyer, Mediator | 75.00% | 82.50% |
| Scientist, Principal | 79.50% | 82.50% |
| Lawyer, Auditor | 74.00% | 81.00% |
| Engineer, Military Commander | 78.00% | 81.00% |
| Ethics Advisor, Lawyer | 75.50% | 79.00% |

Table 3: Evaluation of 30 Role Combinations on SafeEdit evaluated with Qwen3-8B.

# D How Each Benchmark Evaluates

## D.1 Benchmarks

| Benchmark | Evaluator | Metric |
|---|---|---|
| SafeEdit | Fine-tuned RoBERTa-large | Defense Success (DS) |
| SaladBench | Fine-tuned Mistral-7B | Safety Rate (SR) |
| WildJailbreak | Fine-tuned Llama2-13B | Attack Success Rate (ASR) |
| HarmfulQA | GPT-5 | Attack Success Rate (ASR) |
| GSM-Danger | GPT-5 | Attack Success Rate (ASR) |

Table 4: Benchmarks, evaluators, and corresponding metrics used in our evaluation. These methods are proposed by the benchmark themselves, except we changed from GPT-4 to GPT-5 for the last three.

